# Less Is More: Generating Time Series with LLaMA-Style Autoregression in Simple Factorized Latent Spaces

## Abstract

Generative models for multivariate time series are essential for data augmentation, simulation, and privacy preservation, yet current state-of-the-art diffusion-based approaches are slow and limited to fixed-length windows. We propose FAR-TS, a simple yet effective framework that combines disentangled Factorization with an AutoRegressive Transformer over a discrete, quantized latent space to generate Time Series. Each time series is decomposed into a data-adaptive basis that captures static cross-channel correlations and temporal coefficients that are vector-quantized into discrete tokens. A LLaMA-style autoregressive Transformer then models these token sequences, enabling fast and controllable generation of sequences with arbitrary length. Owing to its streamlined design, FAR-TS achieves orders-of-magnitude faster generation than Diffusion-TS while preserving cross-channel correlations and an interpretable latent space, enabling high-quality and flexible time series synthesis.

## 1 Introduction

Multivariate time series data are central to a wide range of real-world applications, including financial markets, energy systems, healthcare monitoring, and sensor networks (Tsay, 2013; Lim & Zohren, 2021; Fang et al., 2024a). Despite their importance, collecting high-quality time series remains challenging due to privacy concerns, acquisition costs, and the prevalence of sparsity or noise in measurements (Gonen et al., 2025). For example, financial institutions often restrict access to trading records due to confidentiality, and sensor networks frequently produce incomplete or corrupted signals. These issues limit the availability of reliable data for model training and evaluation, creating a strong demand for methods that can generate realistic and diverse synthetic time series, particularly under conditions of variable length and high dimensionality.

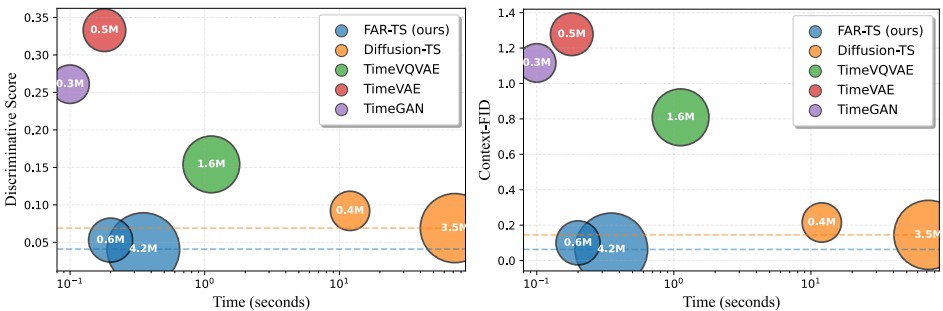

Figure 1: Inference time versus Discriminative Score (left) and Context-FID (right) on the ETTh dataset. For both metrics, lower values indicate better performance, so models closer to the lower-left corner perform best. Bubble size denotes model size, and dashed lines mark the results of the corresponding models. The proposed FAR-TS achieves on average a 50% performance gain with orders-of-magnitude faster generation and shows better scalability than Diffusion-TS.

| Model | Speed | Multivariate | Interpretablity | Length |
|-------|-------|--------------|-----------------|--------|
| TimeGAN (Yoon et al., 2019) | Fast | Limited | Low | Fixed |
| TimeVAE (Desai et al., 2021) | Fast | ✓ | Moderate | Fixed |
| Diffusion-TS (Yuan & Qiao, 2024) | Slow (Iterative) | ✓ | High | Fixed |
| TimeVQVAE (Lee et al., 2023) | Fast | Limited | Moderate | Fixed |
| **FAR-TS (ours)** | **Fast (AR)** | ✓ | **High** | **Arbitrary** |

Table 1: Comparison of representative generative paradigms for time series. FAR-TS uniquely combines interpretability, fast sampling speed, arbitrary-length generation, and a dedicated multivariate design.

The field of time series generation has evolved through several paradigms. Early approaches leveraged generative models such as Variational Autoencoders (VAEs) and Generative Adversarial Networks (GANs) to model time series data (Desai et al., 2021; Yoon et al., 2019). Recently, diffusion-based generative models have emerged as a dominant paradigm, gaining significant attention due to their ability to produce high-quality samples (Yuan & Qiao, 2024). However, these methods suffer from critical limitations that hinder their practical applicability: they are constrained to fixed-length time windows, require iterative denoising processes resulting in slow sampling speeds, and scale poorly with model size.

While autoregressive (AR) architectures, particularly various Transformer variants, have achieved remarkable success in time series forecasting tasks (Nie et al., 2022; Liu et al., 2023b), their potential for generative modeling remains largely unexplored. The tokenize-and-autoregressive paradigm, which has revolutionized natural language processing and image generation, has received limited attention in the time series generation community, despite its potential for fast sampling and arbitrary-length generation. Notably, existing AR methods like TimeVQVAE (Lee et al., 2023) rely on frequency-domain representations for univariate sequences and fail to capture cross-channel relationships, limiting their use for multivariate data and forecasting tasks.

To overcome these limitations, we propose FAR-TS, a simple yet effective framework that integrates the disentangled F̲actorization into the vector quantization (VQ) and A̲R̲ mechanism to generate T̲ime S̲eries. Our approach explicitly decomposes multivariate time series into a shared spatial basis and a set of quantized temporal coefficients. This separation allows the model to capture inter-channel correlations and temporal dynamics through dedicated modules, improving both interpretability and efficiency. The discrete token representation are learned via a LLaMA-style transformer, which enables fast, scalable, and arbitrary-length generation. Additionally, FAR-TS supports introduction of conditional information such as class labels, preserves global and local patterns, and provides a structured latent space that facilitates downstream tasks such as forecasting. A comparison with representative generative paradigms for time series is provided in Table 1.

Our contributions are threefold. First, we introduce a low-rank VQ latent factorization that decomposes multivariate time series into a learnable basis and vector-quantized temporal coefficients, providing an explicit spatiotemporal representation that scales efficiently and supports high-fidelity reconstruction. Second, we design an autoregressive discrete prior with a LLaMA-style Transformer to model token sequences, enabling fast and flexible generation of arbitrary-length sequences with conditional control such as class labels. Third, we conduct experiments on various benchmarks, showing that FAR-TS achieves orders-of-magnitude faster sampling than diffusion-based models while maintaining superior generation and forecasting performance (Figure 1), and further construct a real-world multi-class dataset to assess its conditional generation capabilities.

## 2 BACKGROUND

This section provides the theoretical foundation for our FAR-TS method, focusing on the key technologies that enable our approach: vector quantization combined with autoregressive modeling, and spatiotemporal disentanglement via matrix factorization.

### 2.1 TIME SERIES GENERATION PROBLEM

Given a multivariate time series $X \in \mathbb{R}^{D \times T}$ with $D$ channels and $T$ time steps, our goal is to learn a generative model $p_\theta(X)$ capable of producing realistic and diverse samples, optionally conditioned

on additional information $c$ such as class labels, textual descriptions, or numerical context, i.e., $p_\theta(X \mid c)$. Generating time series is particularly challenging due to long-range temporal dependencies, complex inter-channel correlations, and diverse patterns such as trends, seasonality, and noise. Moreover, the high dimensionality of multivariate sequences requires scalable modeling approaches that preserve both efficiency and generation quality.

## 2.2 Vector Quantization and Autoregressive Modeling

Vector quantization (VQ) provides a powerful mechanism to compress continuous representations into discrete tokens, enabling autoregressive Transformers to directly model long-range dependencies. Given a codebook $\mathcal{E} = \{e_k\}_{k=1}^K$ where $e_k \in \mathbb{R}^R$, VQ maps each continuous vector $v_t \in \mathbb{R}^R$ to the nearest codebook entry:

$$z_t = \arg\min_k \|v_t - e_k\|_2^2. \tag{1}$$

The corresponding quantized vector is then given by $\hat{v}_t = e_{z_t}$. The discrete indices $\{z_t\}$ serve as tokens that can be modeled by powerful sequence models. This approach enables autoregressive generation by factorizing the probability distribution as:

$$p(z_{1:T} \mid c) = \prod_{t=1}^T p(z_t \mid z_{<t}, c), \tag{2}$$

where $c$ is an optional conditioning variable, and the unconditional case is obtained by omitting $c$.

The VQ+AR paradigm originated in computer vision applications, where it revolutionized sequence modeling for structured data (Van Den Oord et al., 2017; Tian et al., 2024; Sun et al., 2024). This combination offers several key advantages: compact representation through discrete tokens, enhanced interpretability through explicit token sequences, support for autoregressive priors, arbitrary-length generation capabilities, and fast inference. While autoregressive Transformers have achieved remarkable success in time series forecasting tasks, their potential for full generative modeling remains largely unexplored.

## 2.3 Disentangle Representation of Time Series

Decomposition is a conventional technique in multivariate time series analysis that separates time series into components capturing distinct patterns, which is useful for exploring complex variations (Cleveland et al., 1990; Brockwell & Davis, 2009). In this work, we employ matrix factorization to explicitly disentangle inter-channel (spatial) and temporal dependencies. For a multivariate time series $X \in \mathbb{R}^{D \times T}$, we decompose it as

$$X = UV^\top + E, \tag{3}$$

where $U \in \mathbb{R}^{D \times R}$ is a basis matrix capturing cross-channel structure, $V \in \mathbb{R}^{T \times R}$ is a temporal coefficient matrix encoding temporal dynamics, and $E \in \mathbb{R}^{D \times T}$ is a residual matrix accounting for noise or fitting errors.

This decomposition principle has a rich history in time series analysis. The seasonal-trend decomposition (STL) framework represents the most classical approach, decomposing univariate series into seasonal, trend, and residual components (Cleveland et al., 1990; BIANCONCINI et al., 2016). Subsequent work has extended this framework to multivariate and probabilistic settings. Recently, groups of advanced time series work (Woo et al., 2022; Liu et al., 2023a; Fang et al., 2024a;b; Deng et al., 2024) have demonstrated that disentangled representations of high-order series achieve significant performance improvements in forecasting and imputation tasks while providing enhanced interpretability.

The factorization approach offers several advantages: improved interpretability through explicit separation of spatial and temporal factors, enhanced model stability by reducing the complexity of joint modeling, scalability for high-dimensional time series, and flexible incorporation of domain priors. This theoretical foundation motivates our design principle of explicit spatiotemporal disentanglement, which we implement through our three-stage pipeline.

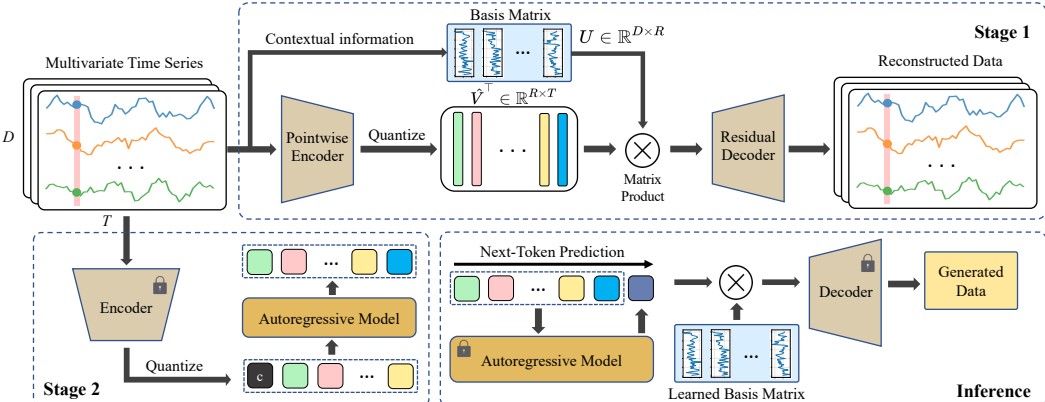

Figure 2: Overview of the FAR-TS pipeline with two training stages and inference. Components marked with a lock are frozen during that stage. **Stage 1**: A pointwise MLP encoder maps each time step of $X \in \mathbb{R}^{T \times D}$ to coefficient vectors, which are quantized into tokens $z$ via a shared codebook. **Stage 2**: A LLaMA-style autoregressive models the token sequence. **Inference**: sampled tokens $\hat{z}$ are mapped to coefficients $\hat{V}$, combined with a learnable spatial basis $U \in \mathbb{R}^{D \times R}$ to reconstruct $\tilde{X}$, and refined by a residual decoder.

## 3 METHOD

Our FAR-TS method is built on a philosophy of principled simplicity. Instead of following the trend of increasingly complex and over-engineered modules, we introduce a streamlined framework that addresses the core challenges of time series generation through a factorized latent space. The key idea is to separate spatial and temporal dependencies: cross-channel correlations are captured in an interpretable latent basis, while complex temporal dynamics are modeled by a powerful autoregressive module. This decoupling yields a model that is efficient, scalable, and interpretable.

The architecture has two main components, trained in a staged manner. First, a spatiotemporal VQ model encodes time series into a factorized latent space, producing discrete temporal tokens disentangled from a learnable spatial basis that captures inter-channel structure. Second, an autoregressive Transformer is trained on the resulting token sequences to learn temporal dynamics of the series. During generation, the Transformer autoregressively samples new token sequences, which are then decoded back into the time series domain using the learned spatial basis and a lightweight decoder. The overall structure of the model is illustrated in Figure 2.

### 3.1 STAGE I: LEARNING A FACTORIZED LATENT SPACE WITH A VQ MODEL

#### 3.1.1 ENCODER: FACTORIZATION INTO SPATIAL BASIS AND TEMPORAL COEFFICIENTS

At the core of FAR-TS is the factorization of a multivariate time series $X \in \mathbb{R}^{T \times D}$ into a spatial basis $U \in \mathbb{R}^{D \times R}$ and a set of temporal coefficients $V \in \mathbb{R}^{T \times R}$. This decomposition separates the modeling of cross-channel correlations, captured by $U$, from temporal dynamics, captured by $V$.

A direct estimation of $U$ and $V$ would require solving matrix inversion or least-squares problems, which is computationally expensive and not easily integrated into end-to-end training. To address this, we adopt an autoencoding formulation that replaces inversion with a neural encoder. Specifically, each input vector $x_t \in \mathbb{R}^D$ is mapped into the coefficient space by a pointwise MLP encoder, $E_\phi : \mathbb{R}^D \to \mathbb{R}^R$, which processes each time step independently:

$$V = E_\phi(X^\top) \in \mathbb{R}^{T \times R}. \tag{4}$$

The reconstruction at each time step is then obtained as $U v_t^\top$, where $v_t$ is the $t$-th row vector of $V$ and $U$ is a shared and learnable basis across all steps.

This design offers several benefits. First, $U$ and $V$ can be jointly optimized via backpropagation, enabling efficient and scalable training without explicit matrix inversion. Second, the encoder is lightweight and interpretable, as it maps each time step independently to a coefficient vector, capturing cross-channel correlations without modeling temporal dependencies. Finally, the basis matrix

provides flexible conditional control: class- or dataset-adaptive bases can be trained to incorporate prior knowledge, improving controllability and adaptability of the generative process.

To further prepare for autoregressive generation, the temporal coefficients are quantized by mapping each $v_t$ to its nearest entry in a learnable codebook $\mathcal{E}$ (see Section 2.2). This yields discrete indices $\{z_t\}_{t=1}^{T}$ and their corresponding quantized representations $\hat{V} = [\hat{v}_1, \ldots, \hat{v}_T]^{\top} \in \mathbb{R}^{T \times R}$.

### 3.1.2 DECODER: RESIDUAL RECONSTRUCTION FROM THE FACTORIZED LATENT SPACE

The decoder reconstructs the signal from the quantized temporal coefficients $\hat{V}$ and the spatial basis $U$. The base reconstruction is obtained by the matrix product

$$\tilde{X} = U\hat{V}^{\top} \in \mathbb{R}^{D \times T}. \tag{5}$$

While quantization simplifies the latent space and facilitates autoregressive modeling, it inevitably introduces discretization errors that may degrade fidelity. To alleviate these artifacts and recover fine-grained local structures, we employ a pointwise refinement decoder $\mathcal{D} : \mathbb{R}^D \to \mathbb{R}^D$ that learns a residual correction. The final reconstruction is given by

$$\hat{X} = \tilde{X} + \mathcal{D}(\tilde{X}) = U\hat{V}^{\top} + \mathcal{D}(U\hat{V}^{\top}), \tag{6}$$

where $\mathcal{D}$ is implemented as a 1D convolutional network. This residual refinement corrects for quantization errors while remaining computationally efficient, effectively approximating the residual matrix in Equation 3 and producing high-quality reconstructions without compromising generation speed.

### 3.1.3 TRAINING OBJECTIVE FOR STAGE I

The VQ stage is trained end-to-end with a loss function combining a reconstruction term with VQ commitment and codebook learning terms. The total loss is:

$$\mathcal{L}_{\text{VQ}} = \underbrace{\|X - \hat{X}\|_F^2}_{\text{Reconstruction}} + \underbrace{\sum_t \left\| \text{sg}[v_t] - \hat{v}_t \right\|_2^2}_{\text{Codebook}} + \beta \underbrace{\sum_t \left\| v_t - \text{sg}[\hat{v}_t] \right\|_2^2}_{\text{Commitment}}, \tag{7}$$

where $\text{sg}[\cdot]$ denotes the stop-gradient operator, and $\beta$ is a hyperparameter.

## 3.2 STAGE II: AUTOREGRESSIVE MODELING IN THE FACTORIZED LATENT SPACE

After learning the factorized latent space, the generation task reduces to modeling the sequence of discrete VQ indices $z = (z_1, \ldots, z_T)$. To this end, we employ a LLaMA-style autoregressive Transformer, which predicts the next token $z_{t+1}$ conditioned on the previous context $z_{\leq t}$. The model is trained to maximize the log-likelihood of the token sequence $p(z_{1:T} \mid c)$ via the standard autoregressive objective defined in Equation 2.

Our architecture follows the LLaMA design (Touvron et al., 2023), which is a decoder-only transformer including pre-normalization via RMSNorm (Zhang & Sennrich, 2019) and rotary positional embeddings (Su et al., 2024). We do not incorporate advanced techinique like AdaLN (Peebles & Xie, 2023) to maintain the standard AR structure used in large language models. The model uses causal attention to respect the temporal ordering of sequences and leverages KV cache (Pope et al., 2023) technique for efficient autoregressive sampling. These design choices make the model highly scalable, capable of handling long token sequences without prohibitive memory or computational costs.

During inference, we perform fast autoregressive decoding with $O(T)$ time complexity. We support various sampling strategies including top-$k$, top-$p$, and temperature-controlled sampling. For class-conditional generation, the class embedding is indexed from a set of learnable embeddings and is used as the prefilling token embedding (Esser et al., 2021). Starting from this token embedding, the model generates the sequence of image tokens by next-token prediction way, and stops at the location of the pre-defined maximum length.

After generating the token sequence $z = (z_1, \ldots, z_T)$, each index is mapped back to its corresponding vector in the codebook $\mathcal{E}$ to obtain the quantized temporal coefficients $\hat{V} = [\hat{v}_1, \ldots, \hat{v}_T]^{\top}$. The

base reconstruction is computed as $U\hat{V}^\top$, which is then refined by the lightweight decoder $\mathcal{D}$ to restore fine-grained details, yielding the final reconstruction $\hat{X} = U\hat{V}^\top + \mathcal{D}(U\hat{V}^\top)$.

Owing to the pointwise design of the VQ decoder and the autoregressive Transformer, FAR-TS naturally supports generation of sequences of arbitrary length. This flexibility enables the model to generate time series of varying lengths without retraining or modifying the architecture, making it suitable for practical scenarios that demand variable-length sequences.

### 3.3 Training Strategy

FAR-TS is trained in a staged manner to simplify optimization and improve stability, as shown in Figure 2. In Stage I, we train the spatiotemporal VQ model to learn the encoder, decoder, spatial basis $U$, and codebook $\mathcal{E}$. Once Stage I converges, we freeze the VQ parameters and pre-compute the discrete token sequences for all training samples, which reduces computational cost and memory usage. In Stage II, we train the LLaMA-style autoregressive Transformer on the pre-computed token sequences. The model learns to predict the next token given the past context. By separating the learning of the latent space and the autoregressive dynamics, this staged approach simplifies optimization, allows modular evaluation of each component, and provides the flexibility to experiment with different AR architectures or conditional settings without retraining the entire model.

### 3.4 Inference Modes

FAR-TS supports multiple inference modes, leveraging its flexible factorized latent space and autoregressive design. For **unconditional generation**, sequences are produced from a beginning-of-sequence (BOS) token or a sample from the prior, allowing flexible synthesis of arbitrary length without conditioning. In **forecasting mode**, observed segments are encoded as token prefixes, and the model autoregressively generates future tokens conditioned on these prefixes, supporting forecasting over varying horizons. For **class-conditional generation**, class information can be directly incorporated into the basis matrix, while temporal information guides the autoregressive generation, enabling precise control over global patterns and style for multi-class synthesis with interpretable and controllable outputs.

### 3.5 Complexity and Efficiency Analysis

Diffusion-based generative models require $S$ iterative denoising steps to produce a single sample, where $S$ is typically in the range of 50 to 1000. Each step involves a forward pass through the full network, making sampling computationally expensive and memory-intensive. In contrast, FAR-TS generates time series autoregressively, producing $T$ tokens in $O(T)$ steps. The decoder-only Transformer leverages causal attention with a KV cache, which avoids recomputation of past hidden states and reduces computational cost. Combined with the lightweight VQ decoder, this design enables FAR-TS to achieve orders-of-magnitude faster generation than diffusion-based models, while preserving high-quality and flexible synthesis (see the scalability analysis in the experimental section).

## 4 Related Work

**GAN- and VAE-based models.** GAN-based methods often use recurrent networks to model temporal dynamics. Mogren (Mogren, 2016) introduced C-RNN-GAN with LSTMs, and RCGAN (Esteban et al., 2017) added label conditioning for medical time series. TimeGAN (Yoon et al., 2019) incorporated an embedding network and supervised loss to better capture temporal dependencies, while RTSGAN (Pei et al., 2021) and PSA-GAN (Jeha et al., 2022) improve quality with progressive growing and self-attention. Due to GAN instabilities, alternative approaches have emerged. Stepwise energy models (Jarrett et al., 2021) imitate sequential behavior via reinforcement learning. Fourier Flows (Alaa et al., 2021) combine normalizing flows with spectral filtering for exact likelihood optimization. TimeVAE (Desai et al., 2021) provides interpretable temporal structure in a VAE, and INRs (Fons et al., 2022) synthesize new sequences via latent embeddings.

**Diffusion-based models.** Diffusion models (Dhariwal & Nichol, 2021) have recently become a leading approach for sequential generation, producing high-fidelity samples by progressively denoising random noise. For time series, methods such as Diffusion-TS (Yuan & Qiao, 2024) and TimeLDM (Qian et al., 2024) achieve state-of-the-art performance. PaD-TS (Li et al., 2025) explores population-level properties, and Naiman (Naiman et al., 2024) transforms time series into images and utilize standard image diffusion modeling. Despite their strong generative capabilities,

their reliance on an iterative sampling process makes them computationally expensive and slow, particularly for long sequences. Furthermore, they are typically trained on fixed-length windows, limiting flexibility for arbitrary-length generation.

**Vector Quantization and Autoregressive Priors.** Vector quantization combined with autoregressive (AR) priors offers a compelling alternative, enabling fast, direct generation in a discrete latent space. In computer vision, models like VQ-GAN (Esser et al., 2021) have shown great success. For time series, TimeVQVAE (Lee et al., 2023) tokenizes univariate frequency-domain representations, while SDformer (Chen et al., 2024) explores improved VQ schemes. While powerful, these methods often do not explicitly model the structure of *multivariate* time series, as their tokenization schemes can entangle spatial and temporal information.

# 5 EXPERIMENTS

## 5.1 EXPERIMENTAL SETUP

**Datasets** We evaluate our model on several widely used multivariate time series datasets. The *ETTh* dataset contains 7 variables of electricity transformer measurements recorded hourly from July 2016 to July 2018. The *ETTm* dataset records the same variables but at a 15-minute frequency. The *fMRI* dataset consists of realistic simulations of blood-oxygen-level-dependent (BOLD) time series. To further assess the ability of our model in generating complex multi-class data, we construct a real-world multi-class sound speed profile (*SSP*) dataset with three distinct classes. Each class corresponds to sound speed data from a different geographical region. Additional details of these datasets and the construction of the SSP dataset are provided in Appendix C.

**Baselines** We select several representative generative models for time series data as benchmarks, including TimeGAN (Yoon et al., 2019), TimeVAE (Desai et al., 2021), Diffusion-TS (Yuan & Qiao, 2024), and TimeVQVAE (Lee et al., 2023). For the prediction task, we additionally include CSDI (Tashiro et al., 2021). For all baselines, we use the official implementations and tune hyperparameters on different dataset. Detailed model settings of FAR-TS is presented in Appendix C.

**Metrics** We evaluate using four standard metrics (Yuan & Qiao, 2024; Jeha et al., 2022): *Discriminative Score*, measuring how well a classifier distinguishes real from synthetic data (lower is better); *Predictive Score*, the MAE of an LSTM trained on synthetic and tested on real data; *Context-FID*, capturing both distributional and temporal similarity; and *Correlational Score*, comparing crosschannel correlations. Together, they assess the quality, usefulness, and temporal consistency of generated time series.

Table 2: Results of unconditional generation with sequence length 48 across different metrics and methods on multiple datasets. Bold indicates the best performance, while underline indicates the second-best performance.

| Metrics | Methods | ETTh | ETTm | fMRI | SSP1 | SSP2 |
|---|---|---|---|---|---|---|
| Context-FID Score | FAR-TS | **0.069±.006** | **0.035±.003** | **0.114±.006** | **0.022±.002** | **0.015±.003** |
| | Diffusion-TS | 0.215±.018 | 0.083±.016 | 0.276±.027 | 0.095±.042 | 0.022±.002 |
| | TimeVQVAE | 0.809±.170 | 12.113±3.67 | 5.616±.299 | 0.133±.109 | 0.308±.297 |
| | TimeGAN | 1.116±.189 | 1.413±.022 | 1.529±.223 | 1.352±.686 | 0.262±.138 |
| | TimeVAE | 1.278±.134 | 0.515±.064 | 23.33±.127 | 0.026±.005 | 0.017±.007 |
| Correlational Score | FAR-TS | **0.048±.012** | **0.024±.004** | **1.653±.030** | **0.073±.034** | **0.056±.014** |
| | Diffusion-TS | 0.071±.003 | 0.034±.002 | 2.059±.014 | 0.123±.022 | 0.085±.016 |
| | TimeVQVAE | 0.191±.022 | 0.773±.044 | 11.420±.047 | 0.607±.062 | 0.180±.010 |
| | TimeGAN | 0.203±.112 | 0.336±.009 | 31.231±.022 | 0.344±.046 | 0.817±.016 |
| | TimeVAE | 0.065±.008 | 0.102±.004 | 16.12±.025 | 0.084±.018 | 0.099±.006 |
| Discriminative Score | FAR-TS | **0.043±.010** | **0.029±.020** | **0.312±.027** | 0.067±.028 | **0.081±.012** |
| | Diffusion-TS | 0.092±.021 | 0.044±.016 | 0.352±.046 | 0.072±.021 | 0.039±.006 |
| | TimeVQVAE | 0.154±.010 | 0.046±.176 | 0.489±.004 | 0.232±.013 | 0.207±.011 |
| | TimeGAN | 0.333±.082 | 0.243±.104 | 0.481±.053 | 0.430±.009 | 0.326±.013 |
| | TimeVAE | 0.261±.074 | 0.064±.089 | 0.434±.086 | **0.043±.016** | 0.084±.023 |
| Predictive Score | FAR-TS | **0.108±.010** | **0.087±.001** | **0.089±.003** | **0.008±.001** | **0.009±.002** |
| | Diffusion-TS | 0.120±.002 | 0.092±.001 | 0.101±.000 | 0.009±.001 | 0.010±.001 |
| | TimeVQVAE | 0.123±.001 | 0.290±.035 | 0.224±.011 | 0.008±.001 | 0.012±.005 |
| | TimeGAN | 0.163±.002 | 0.164±.016 | 0.120±.001 | 0.032±.008 | 0.013±.002 |
| | TimeVAE | 0.130±.003 | 0.112±.004 | 0.103±.001 | 0.009±.001 | 0.012±.002 |

## 5.2 Unconditional Time Series Generation

We first assess FAR-TS on unconditional generation with fixed sequence length, and then analyze its ability to synthesize sequences of arbitrary length.

Table 2 summarizes the results of different methods for 48-length sequence generation, where SSP1 and SSP2 denote the first and second subclass of the SSP dataset, respectively. It can be seen that FAR-TS consistently outperforms all baselines across nearly all evaluation metrics, demonstrating its effectiveness in multivariate time series generation. Notably, it achieves on average 50% lower Context-FID than Diffusion-TS, the state-of-the-art diffusion model, which otherwise delivers the second-best performance in most cases. FAR-TS also significantly surpasses the autoregressive baseline TimeVQVAE, whose poor results reveal the limitations of neglecting cross-channel correlations. In contrast, FAR-TS explicitly disentangles and models both spatial and temporal dependencies through its factorized latent space, allowing it to capture complex variations more effectively. In addition, the decoder-only LLaMA-style Transformer adopted in FAR-TS offers stronger generative capacity than the bidirectional Transformer used in TimeVQVAE, further enhancing performance.

As FAR-TS supports arbitrary-length sequence generation, we evaluate its performance on the ETTh dataset across different lengths. The results are reported in Table 3. While Diffusion-TS and TimeVQVAE are separately trained for each target length, FAR-TS is trained only once with length 48 and directly applied to all other lengths. Despite this, FAR-TS consistently outperforms both baselines across all settings, demonstrating its effectiveness and flexibility in handling variable-length generation.

To further assess synthesis quality, we follow the visualization strategy in Yuan & Qiao (2024), employing both t-SNE and kernel density estimation. Figure 3 shows the results on the ETTh dataset. FAR-TS exhibits closer alignment with the real data distribution, achieving better overlap in the embedding space and more accurate density estimates. Additional visualizations are provided in Appendix B.

Table 3: Results of unconditional generation with multiple length (24, 48, and 96) of ETTh dataset.

| Length | Methods | Context-FID | Correlational Score | Discriminative Score | Predictive Score |
|---|---|---|---|---|---|
| 24 | FAR-TS | **0.045±.005** | **0.044±.008** | **0.035±.002** | **0.106±.008** |
| | DiffusionTS | 0.116±.010 | 0.049±.008 | 0.061±.009 | 0.119±.002 |
| | TimeVQVAE | 1.728±.056 | 0.345±.015 | 0.191±.001 | 0.110±.003 |
| 48 (Training length) | FAR-TS | **0.069±.006** | **0.052±.012** | **0.043±.010** | **0.108±.010** |
| | DiffusionTS | 0.215±.018 | 0.071±.003 | 0.092±.021 | 0.120±.002 |
| | TimeVQVAE | 0.809±.170 | 0.191±.022 | 0.154±.010 | 0.123±.001 |
| 96 | FAR-TS | **0.324±.014** | **0.062±.005** | **0.096±.010** | **0.110±.001** |
| | DiffusionTS | 0.716±.037 | 0.081±.005 | 0.134±.010 | 0.120±.002 |
| | TimeVQVAE | 1.830±.196 | 0.128±.019 | 0.167±.014 | 0.129±.015 |

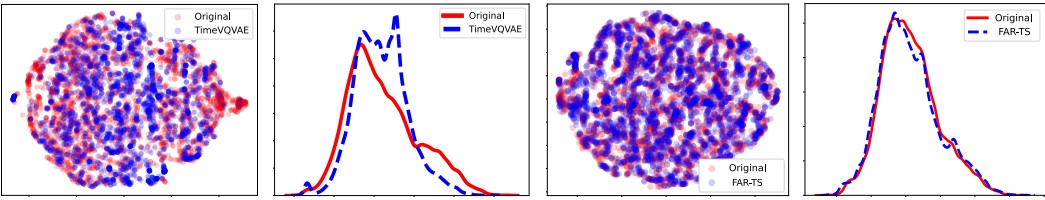

Figure 3: Visualizations of the time series generated by TimeVQVAE and FAR-TS.

## 5.3 Conditional Time Series Generation

We next evaluate the conditional generation performance of different methods, focusing on two tasks: **forecasting** and **multi-class generation**. Forecasting is assessed using Root Mean Squared Error (RMSE) and Mean Absolute Error (MAE). Table 4 reports forecasting results on the SSP dataset across different sequence lengths. For in-window prediction, we use $48-p$ observations to predict a future segment of length $p$. For out-window prediction, the input length is fixed to 24. Since diffusion-based models cannot be directly applied in the out-window setting, we adopt a sliding-window strategy where previously generated segments are fed back for continuation. FAR-TS consistently achieves the best performance across both settings, indicating reliable forecasting

ability. This improvement stems from disentangling spatial and temporal components in the latent space and the autoregressive model's ability to capture temporal dynamics.

We also evaluate **multi-class conditional generation**, where class labels are provided as additional conditions. In FAR-TS, each subclass is assigned a distinct basis matrix, and class information is used to guide autoregressive training. Results are shown in Table 7. Since Diffusion-TS does not support class conditioning, we train separate models for each subclass for comparison. FAR-TS surpasses TimeVQVAE in this setting, highlighting that its factorized design naturally accommodates multi-class generation, whereas TimeVQVAE struggles to model subclass-specific distributions effectively.

Table 4: Forecasting RMSEs/MAEs of different models on SSP data for different length.

| Models | In window | | Out window | | |
| --- | --- | --- | --- | --- | --- |
| | 24 | 36 | 48 | 72 | 96 |
| FAR-TS | **0.023/0.015** | **0.024/0.015** | **0.025/0.016** | **0.026/0.017** | **0.033/0.021** |
| Diffusion–TS | 0.024/0.016 | 0.030/0.021 | 0.027/.019 | 0.034/0.023 | 0.039/0.027 |
| CSDI | 0.091/0.042 | 0.140/0.095 | 0.163/0.131 | 0.175/0.141 | 0.150/0.119 |

## 5.4 SCALABILITY AND INTERPRETABILITY

We first analyze scalability, a key strength of the proposed approach. Figure 4a compares the runtime of FAR-TS and Diffusion-TS across different model sizes, with detailed settings and model size comparisons given in Table 9 and Table 10. Figure 1 and Table 13 further present performance comparisons across methods and model sizes. The results show that FAR-TS not only achieves faster generation but also scales more effectively, as larger models yield improved performance while runtime grows much more slowly. This advantage comes from the factorized latent representation and autoregressive formulation, which together enable efficient training and inference at scale.

We then examine interpretability, which in FAR-TS stems from factorizing time series into a basis matrix and temporal coefficients. The basis captures cross-channel correlations, while the coefficients describe their temporal evolution. Figure 4b shows a factorization example of the fMRI data, where the learned bases effectively captures the variations among different channels. Additional decoder interpretability results and a comparison with dictionary learning are provided in Appendix B.

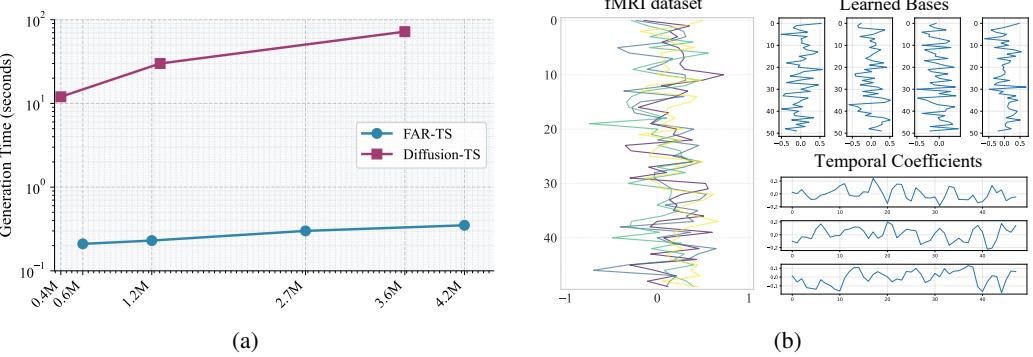

(a)            (b)

Figure 4: (a) Runtime of Diffusion-TS and FAR-TS with different model sizes. (b) Samples from the fMRI dataset, along with the learned basis functions and temporal coefficients of FAR-TS.

## 6 CONCLUSION

We introduced FAR-TS, a framework for multivariate time series generation that combines disentangled factorization with an autoregressive Transformer over a discrete latent space. By disentangling spatial and temporal dependencies via a learnable basis and quantized temporal coefficients, FAR-TS achieves interpretable, controllable, and scalable generation. Experiments on diverse benchmarks show that FAR-TS outperforms existing diffusion- and autoregressive-based methods while providing orders-of-magnitude faster inference. Our approach offers a flexible, efficient, and high-quality solution for generating multivariate time series and supporting downstream tasks such as forecasting and multi-class conditional synthesis.

## ETHICS STATEMENT

Our work focuses on generating synthetic multivariate time series for data augmentation, simulation, and privacy-preserving analysis. All datasets used are publicly available and comply with applicable privacy requirements. FAR-TS does not involve direct interaction with human subjects and does not include mechanisms for identifying individuals. While our model aims to support responsible research, synthetic data could be misused if applied without caution. We encourage careful consideration of intended use and adherence to ethical guidelines. No conflicts of interest or sources of methodological bias are present.

## REPRODUCIBILITY STATEMENT

To ensure reproducibility, we provide detailed descriptions of our model architecture, training procedures, and evaluation metrics in the main text and Appendix C. Additionally, all experiments, including baseline comparisons and ablation studies, are documented with sufficient detail to allow independent replication. We also release the code and scripts to reproduce all results at `https://anonymous.4open.science/r/FAR-TS-1976`.

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

## A    USE OF LARGE LANGUAGE MODELS (LLMS)

In preparing this manuscript, we employed a Large Language Model (LLM) as a general-purpose writing assistant. Specifically, the LLM was used to polish the language, improve clarity and flow, and enhance the presentation of the text. All technical content, experimental design, data analysis, and model development were performed independently by the authors. The LLM was not used to generate any novel scientific ideas, experimental results, or interpretations.

## B    ADDITIONAL GENERATION RESULTS

In this section, we provide supplementary experiments that were omitted from the main body of the paper due to space constraints.

### B.1    GENERATION OF VARIABLE-LENGTH FMRI TIME SERIES

We evaluate the ability of different methods to generate fMRI time series of varying lengths. As reported in Table 5, the proposed model demonstrates both flexibility and effectiveness in generating high-quality sequences of arbitrary length, consistently outperforming the benchmark methods.

Table 5: Comparison of methods across different metrics at varying time lengths (24, 48 and 96) on fMRI dataset. Bold indicates the best performance.

| Length | Methods | Context-FID | Correlational Score | Discriminative Score | Predictive Score |
|--------|---------|-------------|---------------------|----------------------|------------------|
| 24 | FAR-TS | **0.090±.005** | **1.402±.035** | 0.220±.030 | **0.096±.000** |
|    | DiffusionTS | 0.105±.006 | 1.411±.042 | **0.167±.023** | 0.099±.000 |
|    | TimeVQVAE | 3.486±.208 | 13.413±.057 | 0.477±.054 | 0.263±.015 |
| 48 (Training length) | FAR-TS | **0.114±.006** | **1.653±.030** | **0.312±.027** | **0.089±.003** |
|    | DiffusionTS | 0.276±.027 | 2.059±.014 | 0.352±.046 | 0.101±.000 |
|    | TimeVQVAE | 5.616±.299 | 11.420±.047 | 0.489±.004 | 0.224±.011 |
| 96 | FAR-TS | **0.191±.018** | **1.723±.018** | **0.215±.027** | **0.087±.003** |
|    | DiffusionTS | 0.530±.046 | 2.102±.011 | 0.464±.056 | 0.101±.000 |
|    | TimeVQVAE | 11.167±1.243 | 12.245±.034 | 0.351±.186 | 0.200±.004 |

### B.2    ADDITIONAL PLOTS ON ETTH DATASET

To assess how well the generated distributions align with the real data, Figure 5 presents t-SNE visualizations and kernel density estimates for FAR-TS across different sequence lengths. For comparison, the corresponding results of Diffusion-TS and TimeVQVAE are also included. The results indicate that the proposed method achieves closer overlap with the real data distribution and a better alignment of kernel density estimation than the baseline approaches.

### B.3    ADDITIONAL FORECASTING RESULTS

We provide additional forecasting results on the fMRI dataset, using the same experimental settings as in the main text. As shown in Table 6, CSDI achieves the best performance in the in-window setting, which can be explained by its specific design and training strategy. In contrast, FAR-TS demonstrates stronger out-window forecasting ability, benefiting from the extrapolation capability of its autoregressive formulation and VQ design. Furthermore, FAR-TS consistently surpasses Diffusion-TS across most cases, underscoring its effectiveness for downstream tasks such as forecasting.

### B.4    MULTI-CLASS GENERATION RESULTS

The multi-class conditional generation results on the SSP dataset are presented in Table 7.

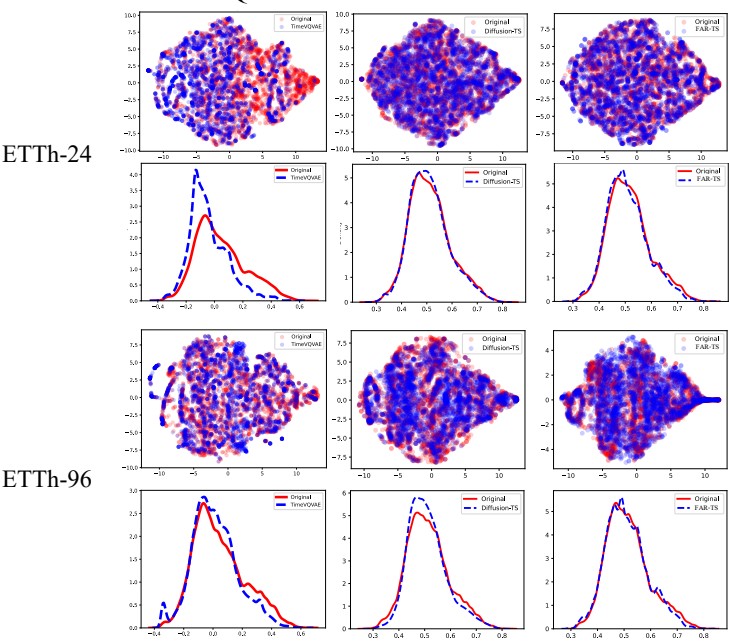

Figure 5: Visualizations of the time series synthesized by FAR-TS, TimeVQVAE, and Diffusion-TS on different length of ETTh.

Table 6: Prediction RMSEs/MAEs of different models on fMRI data under given length.

| Models | In window | | Out window | | |
|--------|-----------|---|------------|---|---|
| | 24 | 36 | 48 | 72 | 96 |
| FAR-TS | 0.172/0.136 | 0.177/0.140 | **0.176/0.138** | **0.175/0.136** | **0.173/0.134** |
| Diffusion–TS | 0.203/0.162 | 0.209/0.167 | 0.215/.173 | 0.223/0.177 | 0.223/0.178 |
| CSDI | **0.147/0.085** | **0.152/0.106** | 0.181/0.144 | 0.183/0.146 | 0.185/0.150 |

## B.5 INTERPRETABILITY ANALYSIS

We evaluate the interpretability of the proposed model through additional experiments on the ETTh dataset. Figure 6 shows the learned basis atoms alongside those obtained from dictionary learning for comparison. Dictionary learning is implemented using the *scikit-learn* library in Python with the same settings as in the fMRI experiment: 256 samples, 50 atoms, sparsity parameter 0.5, and a maximum of 1000 iterations. The learned bases capture the main patterns in the data and closely resemble those from dictionary learning, supporting the interpretability of the model in capturing meaningful distributional structures.

We further examine the decomposition into low-rank and residual components, as defined in Eq. (5). Figure 7 presents the two components for the fMRI dataset. The low-rank component captures the dominant variation patterns, while the residual component models smaller fluctuations. This separation enhances interpretability by distinguishing global structure from fine-scale variations, providing further insight into the model's behavior.

## C MODEL AND DATASET SETTINGS

This section provides detailed information about the experimental setup and datasets to ensure clarity and reproducibility.

Table 7: Multi-class conditional generation results on the SSP dataset.

| Scenario | Methods | Context-FID | Correlational Score | Discriminative Score | Predictive Score |
|---|---|---|---|---|---|
| SSP1 | FAR-TS | **0.028±.002** | **0.076±.034** | **0.057±.028** | **0.008±.001** |
| | DiffusionTS | 0.095±.042 | 0.123±.020 | 0.072±.021 | 0.009±.001 |
| | TimeVQVAE | 0.467±.137 | 0.786±.015 | 0.392±.029 | 0.013±.004 |
| SSP2 | FAR-TS | **0.018±.003** | **0.045±.011** | **0.048±.020** | **0.009±.000** |
| | DiffusionTS | 0.022±.002 | 0.085±.016 | 0.039±.006 | 0.011±.001 |
| | TimeVQVAE | 1.889±.616 | 0.541±.005 | 0.432±.005 | 0.013±.002 |
| SSP3 | FAR-TS | **0.022±.006** | **0.090±.008** | **0.048±.015** | **0.008±.000** |
| | DiffusionTS | 0.089±.018 | 0.093±.003 | 0.055±.007 | 0.009±.000 |
| | TimeVQVAE | 2.903±.655 | 0.871±.074 | 0.305±.021 | 0.024±.027 |

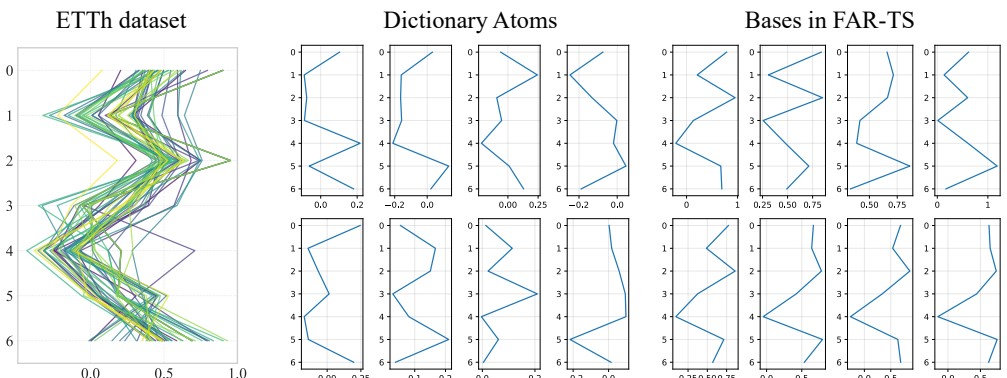

Figure 6: ETTh samples with learned basis atoms from FAR-TS and dictionary learning for comparison.

## C.1 DATASET INFORMATION

Table 8 summarizes the datasets used in our experiments. To evaluate multi-class generation, we construct a real-world SSP dataset following Li et al. (2024). It contains three subclasses, each consisting of sound speed profiles from different geographical regions over the year 2022. Additional details are available in Li et al. (2024). Figure 8 presents examples from the SSP1 data, which show strong correlations among variables due to their spatial proximity. For all datasets, we use 95% for training and 5% for testing.

Table 8: Dataset Details.

| Dataset | # of Samples | dim | Link |
|---|---|---|---|
| ETTh | 17420 | 7 | https://github.com/zhouhaoyi/ETDataset |
| ETTm | 15000 | 7 | https://github.com/zhouhaoyi/ETDataset |
| fMRI | 10000 | 50 | https://www.fmrihub.ox.ac.uk |
| SSP1 | 14360 | 10 | https://github.com/OceanSTARLab/DiffusionSSF |
| SSP2 | 14360 | 10 | https://github.com/OceanSTARLab/DiffusionSSF |
| SSP3 | 14360 | 10 | https://github.com/OceanSTARLab/DiffusionSSF |

## C.2 MODEL CONFIGURATIONS

This subsection describes the model settings for both the proposed FAR-TS and the baseline methods. The hyper-parameters of FAR-TS are listed in Table 9. And the model size are give in Table 10, where we also report the model sizes of the baselines for comparison.

Table 11 lists the training parameters for the VQ and AR models, which are kept consistent across all datasets. For time series generation, we use a sampling temperature of 1.0 with top-$k = 1000$

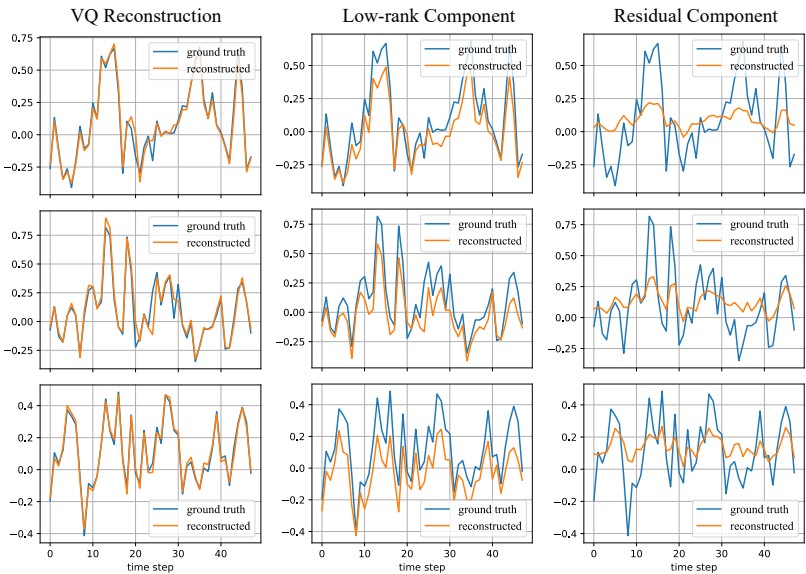

Figure 7: Reconstruction of VQ mode and the low-rank and redisual component, where the first three dimensions are shown for clarity.

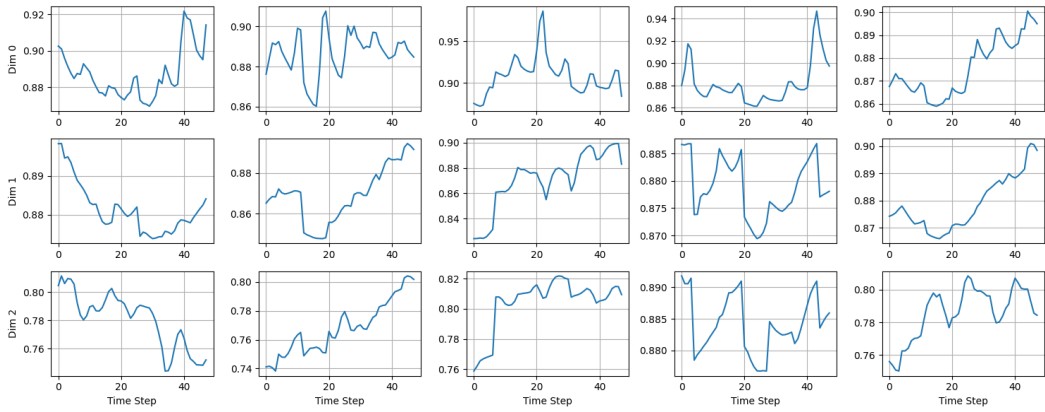

Figure 8: Samples of the SSP1 dataset.

and top-$p = 1.0$ as the default setting. For forecasting, we set the temperature to 0.5 and top-$k = 50$ to reduce randomness and improve stability.

## D  ABLATION STUDY

### D.1  ABLATION OF THE VQ MODEL

We conduct an ablation study to evaluate the contribution of key components in our VQ model. We compare the full model against two variants: (1) w/o matrix factorization, which removes the multiplication with factor bases during training, and (2) w/o residual component, which removes the residual connection in the decoder.

As shown in Table 12, the removal of either component leads to a consistent degradation in performance, as measured by RMSE. This confirms that both the matrix factorization and the residual design are crucial for the model's effectiveness. The factor matrix, in particular, is important not only for encoding prior information and learning interpretable bases but also for achieving superior reconstruction accuracy.

Table 9: Hyperparameters and model architectures of the FAT-TS for different datasets.

| Parameter | ETTh | ETTm | fMRI | SSP1 | SSP2 |
|---|---|---|---|---|---|
| Rank ($R$) | 32 | 32 | 100 | 50 | 50 |
| Codebook Size ($K$) | 4096 | 8192 | 16384 | 16384 | 16384 |
| MLP Max Dim | 2048 | 2048 | 2048 | 2048 | 2048 |
| Commitment Loss ($\beta$) | 0.25 | 0.25 | 0.25 | 0.25 | 0.25 |
| Decoder Channels | 256 | 1024 | 1024 | 1024 | 1024 |
| Model Dimension | 192 | 192 | 256 | 192 | 192 |
| Number of Layers | 6 | 6 | 8 | 6 | 6 |
| Number of Heads | 6 | 6 | 8 | 6 | 6 |
| Dropout Rate | 0.1 | 0.1 | 0.1 | 0.1 | 0.1 |

Table 10: Comparison of Model Size.

| Model | ETTh | fMRI | SSP1 |
|---|---|---|---|
| TimeVAE | 0.50M | 0.50M | 0.50M |
| TimeGAN | 0.33M | 0.33M | 0.33M |
| TimeVQVAE | 1.57M | 1.57M | 1.57M |
| Diffusion-TS | 0.35M | 1.22M | 0.35M |
| FAR-TS | 4.23M | 8.92M | 4.23M |

## D.2 ABLATION STUDY ON AR MODEL SIZE

This subsection presents an ablation study on the impact of model size on the performance of FAR-TS. We evaluate several configurations of our model with varying capacities and compare them against Diffusion-TS. The results, summarized in Table 13, show that FAR-TS consistently out-performs Diffusion-TS across various tested model sizes. Notably, even the smallest configuration of FAR-TS surpasses the baselines, highlighting the efficiency of the factorized representation and autoregressive decoding. As the model size increases, performance further improves in a stable manner, whereas the gains for the baselines are either marginal or achieved at significantly higher computational cost. These results suggest that FAR-TS not only delivers stronger performance but also scales more effectively with capacity, making it better suited for large-scale time series generation tasks.

## D.3 ABLATION STUDY ON GENERATION LENGTH

We further evaluate the impact of sequence length on runtime by comparing FAR-TS with Diffusion-TS on the ETTh dataset. The results are shown in Figure 9. As expected, the runtime of both methods increases with sequence length. However, FAR-TS consistently requires substantially less time across all lengths, demonstrating the scalability of the autoregressive design for generating variable-length sequences.

Table 11: Model Training Parameters.

| Model | Optimizer | LR | Epoch | Beta(Adam) | Batch Size |
|-------|-----------|------|-------|--------------|------------|
| VQ | Adam | 1.00E-04 | 100 | (0.9, 0.999) | 128 |
| AR | Adam | 1.00E-04 | 200 | (0.9, 0.95) | 64 |

Table 12: Ablation study of the VQ model with different model architecture.

| Model | ETTh | fMRI | SSP1 |
|-------|------|------|------|
| VQ | **0.038** | **0.051** | **0.014** |
| VQ w/o matrix U | 0.040 | 0.052 | 0.016 |
| VQ w/o Residual | 0.057 | 0.116 | 0.046 |

Table 13: Comparison of FAR-TS and Diffusion-TS with different sizes across metrics. Bold indicates the best performance.

| Model | Size | Context-FID | Cross Correlation | Discriminative | Predictive |
|-------|------|-------------|-------------------|----------------|------------|
| FAR-TS | 4.22M | **0.069±.006** | **0.048±.012** | **0.043±.010** | **0.108±.010** |
| | 1.45M | 0.076±.006 | 0.058±.009 | 0.046±.010 | 0.110±.008 |
| | 0.57M | 0.082±.008 | 0.064±.010 | 0.053 ±.012 | 0.110±.008 |
| Diffusion-TS | 3.48M | 0.145±.009 | 0.058±.007 | 0.067±.021 | 0.115±.007 |
| | 1.31M | 0.170±.008 | 0.061±.006 | 0.078±.017 | 0.117±.003 |
| | 0.35M | 0.215±.018 | 0.071±.003 | 0.092±.021 | 0.120±.002 |

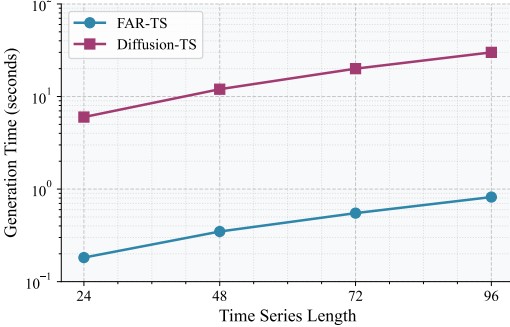

Figure 9: Runtime comparison of FAR-TS and Diffusion-TS with respect to generation length.