# OpenReview forum: "Less Is More: Generating Time Series with LLaMA-Style Autoregression in Simple Factorized Latent Spaces"
_ICLR.cc/2026/Conference — ICLR 2026 Conference Withdrawn Submission_

### Official Review · Reviewer_nuhQ · 2025-10-25

**Soundness:** 3
**Presentation:** 3
**Contribution:** 1
**Rating:** 2
**Confidence:** 5

**Summary:**

This paper proposes FAR-TS, a framework for multivariate time series generation that combines matrix factorization with vector quantization (VQ) and autoregressive (AR) transformers. The approach decomposes time series into a spatial basis matrix and quantized temporal coefficients, then models token sequences using a LLaMA-style transformer. The method claims to achieve faster generation than diffusion models while supporting arbitrary-length sequences and maintaining interpretability.

**Strengths:**

1. Supports arbitrary-length generation, which is valuable for real applications
2. The method achieves orders-of-magnitude faster inference compared to Diffusion-TS
3. Comprehensive experimental evaluation across multiple datasets (ETTh, ETTm, fMRI, SSP) with diverse metrics
4. Extensive ablation studies provided (in the appendix section)
5. Generally well-written with informative visualizations

**Weaknesses:**

1. This paper has limited novelty which combines existing techniques without significant innovations. For example, Matrix factorization for time series is well stablished as well as the Vector quantization
2. The paper compares FAR-TS (4.23M-8.92M parameters) against Diffusion-TS (0.35M-1.22M parameters). This represents a 10-25× larger model, making all performance comparisons fundamentally unfair
3. The paper provides no analysis to verify that generated samples are authentic rather than memorized copies of training data. The extremely low discriminative scores could indicate either successful generation or problematic memorization. So, the authenticity check for the synthetic samples are necessary

**Questions:**

1. Can you provide fair comparisons with matched parameter budget?
2. How do you explain TimeVQVAE's poor performance?
3. How does performance scale to 5-10x training length or higher sequence length to justify arbitrary-length claims?

---

### Official Review · Reviewer_2Njh · 2025-10-29

**Soundness:** 3
**Presentation:** 3
**Contribution:** 3
**Rating:** 6
**Confidence:** 4

**Summary:**

This paper presents FAR-TS, a generative model for multivariate time series that aims to address the limitations of current diffusion-based approaches, namely slow sampling speed and restriction to fixed-length windows. The core idea is to combine a factorized latent space representation (separating a spatial basis capturing cross-channel correlations from temporal coefficients) with vector quantization (VQ) and a LLaMA-style autoregressive (AR) Transformer operating on the discrete temporal tokens. This design enables fast, arbitrary-length generation with potential interpretability benefits.

The work addresses a relevant and challenging problem in time series generation. The proposed combination of factorization, VQ, and a modern AR architecture is interesting and leads to demonstrably faster inference speeds compared to diffusion models, which is a significant practical advantage. The empirical results across several datasets and metrics appear strong, often outperforming existing methods.

However, the paper could be strengthened by clarifying the novelty of its specific components within the established VQ-AR paradigm, providing a more nuanced comparison regarding generation quality (beyond just speed), and offering deeper insights into the factorization process and its implications.

**Strengths:**

1.  **Addresses Key Limitations:** The paper directly tackles significant drawbacks of state-of-the-art diffusion models for time series: slow iterative sampling and fixed window lengths. The proposed AR approach offers a compelling alternative.
2.  **Novel Combination for Time Series:** While VQ and AR Transformers are established techniques, their specific application combined with an explicit *factorized* latent space (spatial basis U + temporal coefficients V) for *multivariate* time series generation appears novel and well-motivated. This contrasts with prior work like TimeVQVAE which focused on univariate frequency-domain representations.
3. **Significant Efficiency Gains:** The AR nature of FAR-TS, leveraging techniques like KV caching, leads to orders-of-magnitude faster generation compared to iterative diffusion models like Diffusion-TS. This is clearly demonstrated empirically (Fig 1, Fig 4a, Fig 9) and represents a major practical contribution.
4. **Flexibility and Controllability:** The AR framework naturally allows for generating sequences of arbitrary length, a key advantage over fixed-window models. The paper also demonstrates flexibility through conditional generation modes like forecasting and multi-class synthesis.
5. **Interpretability Potential:** The explicit factorization into a spatial basis U and temporal coefficients V offers potential interpretability regarding cross-channel structures versus temporal dynamics, as illustrated in the fMRI example (Fig 4b) and basis comparison (Fig 6).
6.  **Strong Empirical Performance:** FAR-TS consistently achieves competitive or superior results compared to strong baselines (TimeGAN, TimeVAE, Diffusion-TS, TimeVQVAE) across multiple standard metrics (Context-FID, Discriminative/Predictive Score, Correlational Score) and datasets.

**Weaknesses:**

1. **Limited Novelty Within Components:** : The overall architecture is coherent but relies on standard elements (VQ, AR Transformer, low-rank factorization). The paper should clarify domain-specific innovations—for example, whether adaptations like RoPE or RMSNorm required modification for quantized temporal tokens, and whether the factorization encoder introduces unique advantages.
2. **Incomplete Discussion of Quality Trade-offs:** : While FAR-TS excels in speed and often in quality, some margins versus diffusion models are small. A nuanced discussion of when AR generation may sacrifice subtle cross-channel correlations or long-range structure would strengthen the empirical claims.
3. **Factorization Sensitivity and Hyperparameter Choice:** : The rank \(R\) of the factorization is critical yet only reported per dataset (e.g., 32/50/100). The paper should analyze how varying \(R\) affects reconstruction error, Context-FID, and generation diversity. Similarly, the fixed dataset-level \(U\) may limit adaptability if correlations evolve over long sequences.
4. **Training Efficiency Discussion Missing:**: Although inference is fast, the two-stage training (factorization + VQ; then AR) may increase overall training time. A comparison of wall-clock training cost with Diffusion-TS or end-to-end approaches would provide a more complete efficiency picture.
5. **Evaluation Scope:**: All metrics are standard (Context-FID, Discriminative, Predictive, Correlational), but additional analyses—e.g., spectral similarity, autocorrelation preservation, or anomaly reconstruction—would better substantiate the generation-quality claims.

**Questions:**

1. Could you elaborate on any specific architectural adaptations made to the LLaMA-style Transformer to handle quantized temporal-coefficient tokens? For example, were RoPE embeddings or normalization layers modified to suit time-series data?
2. How was the rank \(R\) chosen for each dataset, and how sensitive are both the reconstruction and generation performance to this choice?
3. In Table 2, FAR-TS achieves uniformly better Correlational Scores than Diffusion-TS. Could you comment on what factors enable better cross-channel correlation capture despite using a discrete AR model?
4. While inference speedups are clear, how does total training time (Stage I + Stage II) compare to training a diffusion-based model for the same data volume and epochs? Is there a measurable training-efficiency gain or trade-off?

---

### Official Review · Reviewer_9ZM1 · 2025-10-29

**Soundness:** 2
**Presentation:** 3
**Contribution:** 2
**Rating:** 2
**Confidence:** 4

**Summary:**

The author proposes a novel discrete framework FAR-TS with disentangled Factorization and AutoRegressive Transformer for time series generation task, which achieves the competitive performance of efficiency and effectiveness comparisons.

**Strengths:**

1. The paper is easy to follow, the modules are clear.

2. Compared to the diffusion model in time series generation task, FAR-TS shows better generation quality and faster inference speed.

3. The disentangled factorization  strategy in FAR-TS improves the Interpretability of time series data modeling.

4. The experimental results show the  better generation performance, comparing to the popular baselines.

**Weaknesses:**

1. The FAR-TS lacks the novelty in time series generation task, because the Sdformer [1] has shown the effectiveness of VQ- strategy in time series generation task, the only difference is the disentangled Factorization in stage 1. However, the matrix factorization in Eqn.3 shows the last term E, which is not included in VQ-modeling may result in multiple approximations in VQ-strategy to low-quality modeling of stage 1. Can author add the sdformer for comparison ？

2. In Stage2, why the author choose the LLaMA-style transformer for auto regressive generation ？Can author compare the other types of Transformer structure such as vanilla Transformer  to verify the effectiveness of LLaMA-stype ？

3. Some strong baselines are not included in FAR-TS, such as KoVAE [2], Sdformer. Please add

4. According to the table 9 and 10, the memory usage and Codebook Size are much larger than others. Can the author construct a smaller one for comparison ?

[1]. Chen, Zhicheng, et al. "Sdformer: Similarity-driven discrete transformer for time series generation." Advances in Neural Information Processing Systems 37 (2024): 132179-132207.

[2]. Naiman, Ilan, et al. "Generative Modeling of Regular and Irregular Time Series Data via Koopman VAEs." The Twelfth International Conference on Learning Representations.

**Questions:**

Please refer to weaknesses.

---

### Official Review · Reviewer_drXJ · 2025-10-31

**Soundness:** 2
**Presentation:** 2
**Contribution:** 1
**Rating:** 2
**Confidence:** 4

**Summary:**

This paper introduces FAR-TS for generating synthetic time series. The main techniques used in FAR-TS includes vector quantization over multi-variate time series data and autoregressive modeling of temporal patterns. The encoder and decoder for projecting to the codebook are lightweight MLP blocks  and mimics matrix factorization. The model works with arbitrary length. Experiment results demonstrate efficiency at inference time.

**Strengths:**

FAR-TS demonstrates efficiency and scalability over other existing models and supports arbitrary length inference.

The principle of simplicity is well appreciated instead of following the trend of increasingly complex and over-engineered modules.

**Weaknesses:**

The biggest concern I have is with the datasets. Our community really needs to stop using ETT datasets which are nice and smooth and regular and there's no takeaway your can get from evaluating on such datasets. The task diversity is very limited, focusing on clean regular time series. Especially that your length generalization exp picks ETTh.

The experiments are very weak. The number of baselines is very limited, especially that later experiments only compare two baselines.

**Questions:**

what is the sequence length that FAR-TS is trained on?

Are there no more recent models for baselines? Even diffusion-TS is more than a year ago.

It's not clear what in-window and out-window setting is? what 48p to p length is in window and 24 to other lengths is out window?

If the problem motivation of synthetic time series generation is to preserve privacy, qualitative examples in figure 7 do not seem privacy preserving at all.

You mentioned that class information can be directly incorporated into the basis matrix. how exactly? I don't think this is clearly explained.

What is the codebook size and how is it chosen?

---

### Note · Authors · 2025-11-20

I have read and agree with the venue's withdrawal policy on behalf of myself and my co-authors.